# Learning Smooth and Expressive Interatomic Potentials for Physical Property Prediction

Xiang Fu [1]   Brandon M. Wood [1]   Luis Barroso-Luque [1]   Daniel S. Levine [1]   Meng Gao [1]   Misko Dzamba [1]
C. Lawrence Zitnick [1]

## Abstract

Machine learning interatomic potentials (MLIPs) have become increasingly effective at approximating quantum mechanical calculations at a fraction of the computational cost. However, lower errors on held out test sets do not always translate to improved results on downstream physical property prediction tasks. In this paper, we propose testing MLIPs on their practical ability to conserve energy during molecular dynamic simulations. If passed, improved correlations are found between test errors and their performance on physical property prediction tasks. We identify choices which may lead to models failing this test, and use these observations to improve upon highly-expressive models. The resulting model, eSEN, provides state-of-the-art results on a range of physical property prediction tasks, including materials stability prediction, thermal conductivity prediction, and phonon calculations.

## 1. Introduction

Density Functional Theory (DFT), which models the electrons in materials and molecules, serves as the foundation for many modern drug and materials discovery workflows. Unfortunately, DFT calculations are notoriously computationally intensive, scaling cubically with the number of electrons in the system: $O(n^3)$. Machine learning interatomic potentials (MLIPs) are promising in approximating and expediting DFT calculations. With increasing data set sizes and model innovations, MLIPs have shown substantial improvements in accuracy and generalization capabilities (Batatia et al., 2023; Merchant et al., 2023; Yang et al., 2024; Barroso-Luque et al., 2024).

Predicting physical properties in chemistry and materials science often requires complex workflows involving numerous evaluations of DFT or MLIPs. For example, in molecular dynamics (MD) simulations, forces are predicted over thousands to millions of time steps. However, the MLIP literature has mostly focused on assessing models based on energy and force predictions over static DFT test sets rather than directly assessing their performance in complex simulations. This approach has limitations, as improved accuracy on test sets does not always lead to better predictions of physical properties (Póta et al., 2024; Loew et al., 2024).

In this paper, we address two questions: Why does higher test accuracy sometimes fail to enhance a model's ability to predict physical properties, and how can we improve MLIPs to excel in this area? We first outline four critical property prediction tasks and identify the properties required for an MLIP to succeed in these tasks. These properties entail learning a conservative model with continuous and bounded energy derivatives, indicating a smoothly-varying and physically meaningful energy landscape. To test whether these properties hold, we propose testing the ability of MLIPs to practically conserve energy in MD simulations. We demonstrate models that pass this test have a higher correlation between test errors and property prediction accuracy.

Building on these insights, we present a novel MLIP called eSEN and training approach that achieves state-of-the-art (SOTA) performance on complex property prediction tasks. Specifically, our model is capable of running energy-conserving MD simulations for out-of-distribution systems (Figure 1 (a)). For materials stability prediction, eSEN achieves a leading F1 score of 0.831 and a $\kappa_{\text{SRME}}$ of 0.340 on the compliant Matbench-Discovery benchmark (Riebesell et al., 2023; Póta et al., 2024). Previous models are only able to excel in one of these metrics ( Figure 1 (b,c)). We also achieve a SOTA F1 score of 0.925 and $\kappa_{\text{SRME}}$ of 0.170 on the non-compliant category. On the MDR Phonon benchmark (Loew et al., 2024), SOTA results are found (Figure 1 (d)). Finally, eSEN achieves the highest test accuracy on the SPICE-MACE-OFF dataset (Kovács et al., 2023).

---

[1]Fundamental AI Research (FAIR) at Meta. Correspondence to: Xiang Fu <xiangfu@meta.com>, C. Lawrence Zitnick <zitnick@meta.com>.

*Proceedings of the $42^{nd}$ International Conference on Machine Learning*, Vancouver, Canada. PMLR 267, 2025. Copyright 2025 by the author(s).

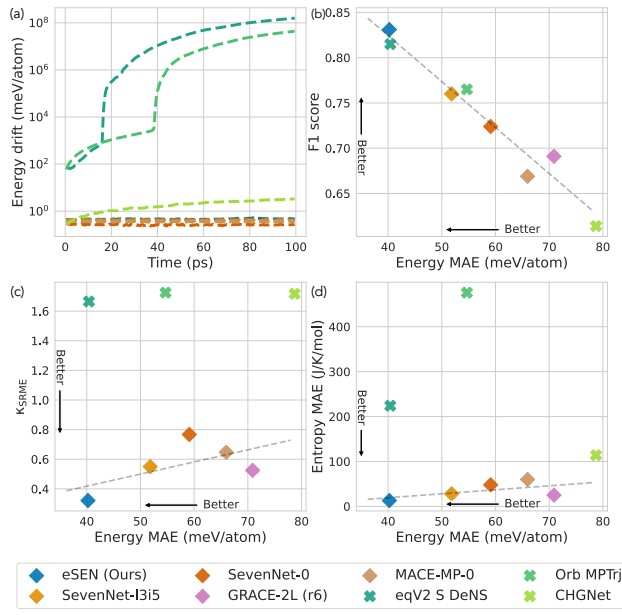

*Figure 1.* (a) Energy conservation in MD simulations. Direct-force models (Orb, eqV2) and CHGNet fail to conserve. (b) A higher F1 score on the Matbench-Discovery strongly correlates with a lower test-set energy MAE. (c) Test-set energy MAE and $\kappa_{\mathrm{SRME}}$ on the Matbench-Discovery benchmark. (d) Test-set energy MAE and vibrational entropy MAE on the MDR Phonon benchmark. Our model (eSEN) achieves the best performance on all benchmarks. A higher correlation between test-set energy MAE and physical property prediction performance can be observed among energy-conserving models. All models are trained on MPTrj.

## 2. Preliminaries

### 2.1. Machine learning interatomic potentials

Under the Born-Oppenheimer approximation (Oppenheimer, 1927) utilized by DFT (Parr et al., 1979), the Potential Energy Surface (PES) can be written as a function of positions, $r$, and atomic numbers, $a$: $E(r, a)$. Per-atom forces can be calculated by taking the negative gradient of the PES with respect to the atom positions, $F = -\nabla_r E$. For periodic systems such as inorganic materials, the lattice parameters $l$ are also considered ($E(r, a, l)$), and the stress $\sigma$ may also be calculated, which can be understood as the gradient of the potential energy surface with respect to the lattice parameters.

The goal of an MLIP (Unke et al., 2021b) is to predict the exact same properties as DFT from a training dataset of DFT calculations (Chanussot et al., 2021; Riebesell et al., 2023; Loew et al., 2024). The most straightforward benchmark for MLIPs is to evaluate the model on a held-out test set of DFT calculations, and compare models based on the mean absolute error (MAE) or root mean squared error (RMSE) of

energies, forces, or stresses. To bridge the gap between these performance metrics and practical applicability, we need to ensure they correlate with physical property prediction tasks, such as those described next.

### 2.2. Physical property prediction tasks

**Geometry optimization/relaxation.** Many computational chemistry and materials science tasks rely on atomic systems being in stable configurations, which correspond to minima of the PES. Stable states are found by minimizing the potential energy using an optimization procedure that iteratively updates atom positions based on the predicted forces ($F = -\nabla_r E$). Given that many physical properties are evaluated at or near equilibrium states, geometry optimization (also referred to as "relaxation") is usually the first step in most computational workflows.

**MD simulations.** Simulating the time evolution of atomic systems enables us to gain understanding of various chemical and biological processes, as well as enabling the calculation of macroscopic properties, such as liquid densities, that can be experimentally verified. For the task of molecular dynamics simulation, we typically use a potential to compute the per-atom forces which are then used to numerically integrate Newton's equations of motion. In this work, we will focus on the **microcanonical ensemble (NVE)**, where the number of particles (N), the volume of the system (V), and the energy of the system (E) are kept constant.

**Phonon and thermal conductivity calculations.** Precise predictions of phonon band structures and vibrational modes are essential for understanding various material properties, including dynamical stability, thermal stability (Bartel, 2022; Fultz, 2010), thermal conductivity (Razeghi, 2002), and optoelectronic behavior (Ganose et al., 2021). The calculation of phonon band structures requires the MLIP to accurately predict higher-order derivatives and capture the subtle curvature of the true PES around critical points. Recent work (Póta et al., 2024) has demonstrated the usage of MLIPs in predicting thermal conductivity ($\kappa$) by solving the Wigner transport equation (Simoncelli et al., 2022). In order to accurately predict $\kappa$, MLIPs must reliably capture both harmonic and anharmonic phonon behavior, which necessitates the calculation of second and third derivatives of the learned PES.

## 3. Desideratum for physical property prediction

We begin the section by defining what it means for an MLIP to be energy conserving, which is a fundamental principle for applications such as MD simulations (Tuckerman, 2023). For many physical property prediction tasks that probe the higher-order derivatives of the PES it is also important that

the PES's derivatives are well-behaved (they exist and are bounded). To indicate whether a PES meets these criteria, we discuss how an MLIP's ability to conserve energy given fixed simulation settings may be used.

### 3.1. Conservative forces

For a force model to be conservative, the work done by moving in a closed path must be zero, i.e., the integration of the forces along any path that starts and ends at the same point is zero:

$$\oint \boldsymbol{F} \cdot d\boldsymbol{r} = 0 \quad (1)$$

This property holds if the forces are calculated as the negative derivative of the PES with respect to the atom positions (Unke et al., 2021b). However, predicting forces as derivatives requires an additional backpropagation step through the network, which increases the computational cost of the MLIP. Alternatively, some networks (Liao et al., 2023; Neumann et al., 2024) directly predict forces using a separate force head to increase efficiency[1]. Although direct-force models can achieve high accuracy, their non-conservative nature leads to significantly larger errors in certain property prediction tasks (Fu et al., 2023; Loew et al., 2024; Póta et al., 2024; Bigi et al., 2024).

### 3.2. Bounded energy derivatives

Conservative forces is a necessary but not sufficient condition for an MLIP to demonstrate energy conservation in MD. In practice, MD simulations use a finite-order numerical integration algorithm and a finite time step $\Delta t$, which introduces truncation errors. The most commonly used integrator for the NVE ensemble is the Verlet algorithm–a second-order integrator. The Verlet integrator is known to approximately conserve the total energy of the system in long-time simulations. As shown by Theorem 5.1 of Hairer et al. 2003, the total energy drift of a simulation satisfies

$$|E(\boldsymbol{r}_T, \boldsymbol{a}) - E(\boldsymbol{r}_0, \boldsymbol{a})| \leq C\Delta t^2 + C_N \Delta t^N T, \quad (2)$$

where $T, 0 \leq T \leq \Delta t^{-N}$, is the total simulation time, $N$ is a positive integer representing the highest order for which the $N$th-order derivative of $E$ is continuously differentiable with a bounded derivative, and $\boldsymbol{r}_0$ and $\boldsymbol{r}_T$ are the starting and ending positions of the atoms in the simulation respectively. The constants $C$ and $C_N$ are independent of $T$ and $\Delta t$. The energy drift bound contains two terms: the first term represents a time-independent fluctuation of $O(\Delta t^2)$, and the second term represents the long-term energy conservation. The proof for this theorem is long and technical, for

---

[1]Strictly speaking, direct-force models are not truly "potentials", but rather (non-conservative) "force fields".

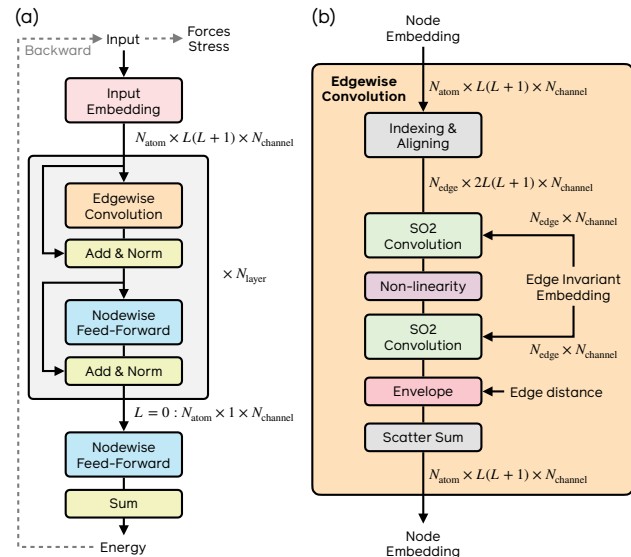

*Figure 2.* (a) The eSEN architecture. The high-level architecture is similar to Transformer/Equiformer, while the edgewise/nodewise layers are simplified/enhanced. The final-layer $L = 0$ features are used to predict nodewise energy, which is summed to get the total potential energy $E$. Forces and stress are obtained through back-propagation. (b) The **Edgewise Convolution** layer in eSEN.

which we refer interested readers to Hairer et al. 2003 and Hairer et al. 2006 for more details.

In Equation (2), the $\Delta t^N$ in the second term and the bound on the simulation time $T \leq \Delta t^{-N}$ implies that the PES must be continuously differentiable to high order for energy conservation in long-time simulations. The critical constant $C_N$ depends on the bounds of the derivatives of $E$ up to the $(N + 1)$th order. This implies that, given a fixed time step size, $E$ and its higher-order derivatives up to the $(N)$th order all need to be continuously differentiable with bounded derivatives to maintain long-time conservation. If the derivatives of a PES are more tightly bound, approximate energy conservation will be maintained even at larger step sizes $\Delta t$. Therefore, the magnitude of $\Delta t$ for which the energy is stable can be viewed as a proxy for the derivative bounds of the estimated PES. Alternatively, if a certain time step is known to be stable when using DFT, we can determine whether an MLIP has similar bounds on higher-order derivatives by testing whether it is also stable using the same time step.

## 4. eSEN

We propose **e**quivariant **S**mooth **E**nergy **N**etwork (**eSEN**), a new MLIP architecture that improves upon architectures that demonstrate high test accuracies to achieve effective physical property predictions. eSEN is a message-passing neural network that conducts multiple blocks of edgewise

and nodewise neural processing. Initially, all nodes are embedded as multi-channel spherical harmonic representations. Each eSEN layer block updates the node embedding by conducting an edgewise convolution, followed by a nodewise feed-forward network with normalization layers and residual connections between all layers.

A model diagram is shown in Figure 2. eSEN utilizes the same SO2 convolution layer from the equivariant spherical channel network (eSCN) architecture (Passaro & Zitnick, 2023) inside the edgewise convolution block. Compared to eSCN, our edgewise convolution blocks first concatenate the source and target node embedding, then apply two SO2 convolution layers with an intermediate non-linearity. We also add an envelope function (details in Section 5) which is not in eSCN. The nodewise feed-forward layer uses two equivariant linear layers and an intermediate SiLU-based gated non-linearity (Weiler et al., 2018; Geiger & Smidt, 2022), which is the same as Equiformer (Liao & Smidt, 2022). Unlike eSCN and EquiformerV2 (Liao et al., 2023), which projects the spherical-harmonics channels onto spatial grids for nodewise processing, the nodewise layers in eSEN do not discretize the node representations. As we demonstrate in Section 5, this design improves the ability of the model to conserve energy. Normalization is performed using the equivariant layer normalization (Ba, 2016) proposed by Equiformer (Liao & Smidt, 2022). In the next section, we conduct an in-depth analysis of the key design choices for energy conservation, which we argue is important for accurate physical property prediction.

# 5. Design choices for enhancing physical property prediction

As discussed in Section 3, having conservative forces with continuous and bounded energy derivatives are properties an MLIP should obey for MD simulations. It can also be seen as a prerequisite for the MLIP to accurately capture higher-order behavior of the PES and thus high accuracy in physical property prediction tasks such as phonon calculations. Motivated by this observation, we identify design choices that impact a model's ability to conserve energy and whether its PES varies smoothly. These design choices can be categorized into three aspects: (1) conservative vs. direct-force prediction; (2) discretization of the representation; and (3) obtaining a continuous and smoothly varying PES. For many of these design aspects, their impact on the desired properties is not well understood.

To quantify whether an MLIP's PES is continuous and smoothly varying, we measure the ability of the resulting MLIP to conserve energy during MD simulations with a predetermined fixed time step. We trained eSEN models under the same hyperparameters while ablating one design choice at a time. We construct out-of-distribution (OOD) MD

simulation tasks for both inorganic materials and organic molecules using models trained on the MPTrj (Jain et al., 2013; Deng et al., 2023) and the SPICE-MACE-OFF (Eastman et al., 2023; Kovács et al., 2023) datasets. For inorganic materials, we compute an average conservation error over 81 NVE MD simulations of 100 ps based on the TM23 dataset's simulation settings (Owen et al., 2024). For organic molecules, we compute an average conservation error over 7 NVE MD simulations of 100 ps based on the MD22 dataset's simulation settings (Chmiela et al., 2023). All eSEN models are 2-layer with 3.2M trainable parameters. We include details regarding the task protocol in Appendix A.

## 5.1. Direct-force prediction

Models that directly predict forces $\hat{F}$ from the atomic configuration may produce forces that are inconsistent with the energy prediction, i.e., $\hat{F} \neq -\nabla_r \hat{E}$, and more importantly are unlikely to be conservative. From the perspective of minimizing the test error, the direct-force approach has strong motivations: it avoids the backward pass for force prediction, which significantly improves model efficiency and enables low-precision training which further accelerates training. Empirically, current SOTA accuracy on the OC20, OC22, and Matbench-Discovery (Chanussot et al., 2021; Tran et al., 2023; Riebesell et al., 2023) benchmarks are achieved by direct-force models. Despite this, the direct-force formulation results in significant energy drift in MD simulations, as shown in Figure 4 (a1, a2). For this reason, we compute forces as the negative gradient of the PES with respect to the atom positions in eSEN.

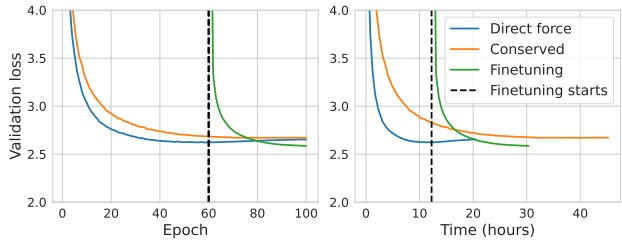

*Figure 3.* Validation loss curves for epoch and wallclock time.

**Direct-force pre-training.** Although direct-force models are not suitable for certain physical property prediction tasks, they may still offer advantages (Bigi et al., 2024; Amin et al., 2025). We demonstrate their efficiency can offer significant benefit as a pre-training strategy for a conservative model. Figure 3 shows the validation loss of 2-layer eSEN models trained on the MPTrj dataset: direct-force, conservative, and conservative fine-tuning from a pre-trained direct-force backbone. We start from a direct-force model trained for 60 epochs, remove its direct-force prediction head, and fine-

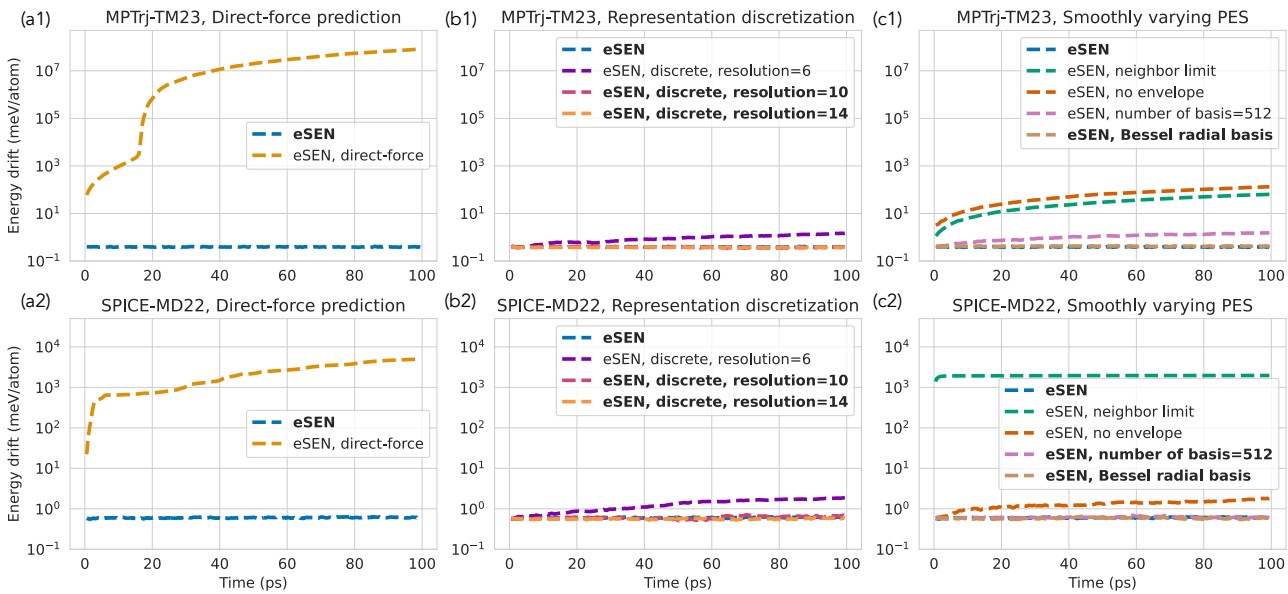

*Figure 4.* Conservation error on the TM23 task (top row) and MD22 task (bottom row) for ablating design choices of eSEN. Models that conserve energy are **bolded** in the legends.

tune using conservative force prediction. The conservative fine-tuned model achieves a lower validation loss after being trained for 40 epochs compared to the from-scratch conservative model being trained for 100 epochs. The fine-tuning strategy also reduces the wallclock time for model training by 40%. The strategy of combining direct-force pre-training and conservative fine-tuning is also shown to be effective under different data/model settings (Bigi et al., 2024).

### 5.2. Representation discretization

As proposed by Cohen & Welling 2016 and later used in eSCN and EquiformerV2 (Zitnick et al., 2022; Passaro & Zitnick, 2023; Liao et al., 2023), non-linearities may be performed by projecting the spherical harmonics to a discrete grid. A $1 \times 1$ convolution or pointwise non-linearity may then be applied to this grid, which then get projected back to the spherical-harmonics space. The non-linear step may introduce higher-frequency signals than cannot be properly represented by the spherical harmonics, i.e., they are beyond the Nyquist frequency. This can lead to sampling errors that break strict equivariance and energy conservation. This problem can be mitigated by sampling the grid at higher resolutions as shown in Figure 4 (b1, b2). In eSEN, we instead use the SiLU-based equivariant Gated non-linearity (Weiler et al., 2018; Geiger & Smidt, 2022) that performs the non-linearity directly in the spherical harmonic representation. This does not require a projection to a discrete grid, so the model is perfectly equivariant and conservative up to numerical accuracy.

### 5.3. Smoothly varying PES

Subtle choices in the design of MLIPs can have a significant impact on whether a PES varies smoothly and can even lead to the presence of discontinuities. These include how neighboring atoms are chosen, whether envelope functions are used near atom distance cutoffs, and which basis functions are used to embed pairwise atom distances. We discuss each of these in turn.

**A maximum number of neighbors limit** in graph construction has been found to improve training efficiency without compromising test error (Liao et al., 2023; Qu & Krishnapriyan, 2024). However, it results in a discontinuity in the learned PES as the nearest-K neighbors may change drastically under a small perturbation of the atom positions. As shown in Figure 4 (c1, c2), having a maximum neighbor limit breaks energy conservation. In eSEN, instead of limiting the number of neighbors, we use the common approach of applying a distance cutoff (6Å) under which all neighbors are kept.

**Envelope functions** were first introduced in the DimeNet architecture (Gasteiger et al., 2020a) to improve model smoothness. The radial basis function used in MLIPs is not twice continuously differentiable due to the use of a finite cutoff during graph construction. By applying a polynomial envelope function on the edge messages, the values in an edge message and its first/higher-order derivatives with respect to atom positions decays to 0 when the edge distance approaches the cutoff distance. Figure 4 (c1, c2)

*Table 1.* Test set MAE for design choices studied in Section 3. The conserved model significantly outperforms the direct-force model on SPICE-MACE-OFF. $N_{\text{basis}} = 512$ performs slightly better on SPICE but slightly worse on MPTrj. Other eSEN variants all have similar test errors on MPTrj/SPICE-MACE-OFF. Energy MAE is in meV/atom. Force MAE is in meV/Å. Stress MAE is in meV/Å/atom.

| Model | MPTrj | | | SPICE | |
|---|---|---|---|---|---|
| | Energy | Force | Stress | Energy | Force |
| eSEN | 17.02 | 43.96 | 0.14 | 0.23 | 6.36 |
| eSEN, direct | 18.66 | 43.62 | 0.16 | 0.56 | 10.98 |
| eSEN, neighbor limit | 17.30 | 44.11 | 0.14 | 0.24 | 6.52 |
| eSEN, no envelope | 17.60 | 44.69 | 0.14 | 0.23 | 6.33 |
| eSEN, $N_{\text{basis}} = 512$ | 19.87 | 48.29 | 0.15 | 0.19 | 5.40 |
| eSEN, Bessel | 17.65 | 44.83 | 0.15 | 0.20 | 5.54 |
| eSEN, discrete, res=6 | 17.05 | 43.10 | 0.14 | 0.26 | 6.34 |
| eSEN, discrete, res=10 | 17.11 | 43.13 | 0.14 | 0.33 | 6.57 |
| eSEN, discrete, res=14 | 17.12 | 43.09 | 0.14 | 0.33 | 6.51 |

shows a model fails to conserve energy without the envelope function.

**Radial basis functions** are commonly used to embed interatomic distances(Bartók et al., 2013). A larger number of basis functions (512 in Passaro & Zitnick 2023, as opposed to 10 in eSEN's default setting) allows higher-frequency signals to pass through the network. This can lead to the PES being more sensitive to small shifts in the atom positions. In our experiments, using a large number of basis functions breaks conservation for the TM23 tasks, but is able to conserve energy for the MD22 tasks. Using a Bessel radial basis function (as opposed to a Gaussian radial basis in the default setting) does not impact conservation properties in both tasks.

### 5.4. Ablation studies

Many of the architecture choices described above have negligible impact on the test set errors as shown in Table 1. However, as shown in Figure 4, they can have a dramatic impact on whether a model is conservative in practice. If a model is found to be conservative, stronger correlations are found between test errors and property prediction tasks (Figure 1 and Figure 6).

## 6. Experiments

In the previous section, we demonstrated the design of eSEN results in its ability to be energy-conserving in MD simulations. In this section, we evaluate eSEN in physical property prediction tasks: (1) materials stability prediction based on geometry optimization; (2) thermal conductivity prediction; and (3) phonon calculation. We also demonstrate the correlation between test energy MAE and physical property prediction tasks for eSEN.

### 6.1. Matbench Discovery

**The Matbench-Discovery benchmark** evaluates a model's ability to predict ground-state (0 K) thermodynamic stability through geometry optimization and energy prediction. It is a widely used benchmark for evaluating ML models in materials discovery. The compliant benchmark only includes models trained on the MPTrj (Jain et al., 2013; Deng et al., 2023) dataset or its subset, which facilitate a fair comparison of model architectures. The F1 score is the primary metric used to rank models. We train an eSEN with 30M parameters on MPTrj for 60 epochs of direct-force pre-training and 40 epochs of conservative fine-tuning. DeNS (Liao et al., 2024) is used during direct-force pre-training. As shown in Table 2, eSEN-30M-MP achieves an F1 score of 0.831—the highest among all compliant models. eSEN-30M-MP also achieves the lowest root mean square deviation (RMSD) when comparing the relaxed structures to the ground truth DFT reference.

**The thermal conductivity prediction task** requires accurate modeling of harmonic and anharmonic phonons in materials, which tests the accuracy of second and third order derivatives of the learned PES. The primary metric is the symmetric relative mean error in predicting thermal conductivity ($\kappa_{\text{SRME}}$). We follow the protocol set forth in the Matbench-Discovery benchmark (Riebesell et al., 2023; Póta et al., 2024) to predict thermal conductivity $\kappa$. After running a structural relaxation, $\kappa$ is computed using second and third order force constants obtained from phonon calculations using the supercell method.

As shown in Table 2, our model achieves a $\kappa_{\text{SRME}}$ of 0.340 under the default evaluation protocol proposed by Póta et al. 2024. Notably, our model excels in both the F1 score and $\kappa_{\text{SRME}}$, while all previous models only achieve SOTA performance on one or the other of these metrics.

**The non-compliant Matbench-Discovery benchmark** includes models trained on datasets other than MPTrj. eSEN-30M-OAM is an eSEN model with 30 million parameters pre-trained on the OMat24 (Barroso-Luque et al., 2024) dataset then fine-tuned on the subsampled Alexandria (sAlex) dataset (Barroso-Luque et al., 2024; Schmidt et al., 2024) and MPTrj dataset. As shown in Table 3, eSEN-30M-OAM achieves the best performance among all non-compliant models with an F1 score of 0.925, a $\kappa_{\text{SRME}}$ of 0.170, and an RMSD of 0.0608, significantly advancing state-of-the-art.

### 6.2. MDR phonon benchmark

The MDR Phonon benchmark (Loew et al., 2024) assesses the performance of MLIPs in predicting key phonon properties, including maximum phonon frequency ($\omega_{\text{max}}$), entropy ($S$), free energy ($F$) and heat capacity at constant volume

*Table 2.* Matbench-Discovery benchmark results of compliant models (trained only on MPtrj or its subset) with results on the unique prototype split. MAE is in units of eV/atom. (↑/↓) stands for higher/lower the better.

| Metric | eSEN-30M-MP | eqV2 S DeNS | MatRIS-MP | AlphaNet-MP | DPA3-v2-MP | ORB v2 MPtrj | SevenNet-l3i5 | GRACE-2L-MPtrj | MACE-MP-0 | CHGNet | M3GNet |
|---|---|---|---|---|---|---|---|---|---|---|---|
| F1 ↑ | **0.831** | 0.815 | 0.809 | 0.799 | 0.786 | 0.765 | 0.760 | 0.691 | 0.669 | 0.613 | 0.569 |
| DAF ↑ | **5.260** | 5.042 | 5.049 | 4.863 | 4.822 | 4.702 | 4.629 | 4.163 | 3.777 | 3.361 | 2.882 |
| Precision ↑ | **0.804** | 0.771 | 0.772 | 0.743 | 0.737 | 0.719 | 0.708 | 0.636 | 0.577 | 0.514 | 0.441 |
| Accuracy ↑ | **0.946** | 0.941 | 0.938 | 0.933 | 0.929 | 0.922 | 0.920 | 0.896 | 0.878 | 0.851 | 0.813 |
| MAE ↓ | **0.033** | 0.036 | 0.037 | 0.041 | 0.039 | 0.045 | 0.044 | 0.052 | 0.057 | 0.063 | 0.075 |
| R2 ↑ | **0.822** | 0.788 | 0.803 | 0.745 | 0.804 | 0.756 | 0.776 | 0.741 | 0.697 | 0.689 | 0.585 |
| $\kappa_{SRME}$ ↓ | **0.340** | 1.676 | 0.861 | 1.31 | 0.959 | 1.725 | 0.550 | 0.525 | 0.647 | 1.717 | 1.412 |
| RMSD ↓ | **0.0752** | 0.0757 | 0.0773 | 0.1067 | 0.0823 | 0.1007 | 0.0847 | 0.0897 | 0.0915 | 0.0949 | 0.1117 |

*Table 3.* Matbench-Discovery benchmark results of non-compliant models with results on the unique prototype split.

| Model | eSEN-30M-OAM | eqV2-M-OAM | ORB v3 | SevenNet-MF-ompa | DPA3-v2-OpenLAM | GRACE-2L-OAM | MatterSim-v1-5M | MACE-MPA-0 | GNoME |
|---|---|---|---|---|---|---|---|---|---|
| F1 ↑ | **0.925** | 0.917 | 0.905 | 0.901 | 0.890 | 0.880 | 0.862 | 0.852 | 0.829 |
| DAF ↑ | **6.069** | 6.047 | 5.912 | 5.825 | 5.747 | 5.774 | 5.852 | 5.582 | 5.523 |
| Precision ↑ | **0.928** | 0.924 | 0.904 | 0.879 | 0.879 | 0.883 | 0.895 | 0.853 | 0.844 |
| Accuracy ↑ | **0.977** | 0.975 | 0.971 | 0.969 | 0.966 | 0.963 | 0.959 | 0.954 | 0.955 |
| MAE ↓ | **0.018** | 0.020 | 0.024 | 0.021 | 0.022 | 0.023 | 0.024 | 0.028 | 0.035 |
| R2 ↑ | 0.866 | 0.848 | 0.821 | 0.867 | **0.869** | 0.862 | 0.863 | 0.842 | 0.785 |
| $\kappa_{SRME}$ ↓ | **0.170** | 1.771 | 0.210 | 0.317 | 0.687 | 0.294 | 0.574 | 0.412 | N/A |
| RMSD ↓ | **0.0608** | 0.0691 | 0.0750 | 0.0639 | 0.0679 | 0.0666 | 0.0733 | 0.0731 | N/A |

($C_V$), for around 10,000 materials. The evaluation follows the testing protocol outlined by Loew et al. 2024. Section 6.2 shows the resulting MAE of our model and those of several other models[2]. eSEN achieves SOTA results in both compliant and non-compliant categories.

Our results are consistent with those reported by Loew et al. 2024, showing that conservative MLIPs significantly outperform direct-force models in terms of prediction accuracy when tested using phonon calculations with a displacement of 0.01 Å. The high error of direct-force models can be largely attributed to high-frequency prediction errors at small displacements (Loew et al., 2024). Increasing the displacement used in the finite-difference phonon calculations to 0.2 Å can considerably improve prediction accuracy of direct-force models (with caveats). We include a more detailed analysis of the relationship between atom displacement and phonon prediction in Appendix B.

In physical phonon calculations, we expect the results to converge as the displacement goes to zero. By examining the resulting phonon band structure, we can gain insight into this behavior. Figure 5 presents the predicted phonon band structure and density of states for three representative materials using eSEN. The predicted phonon bands exhibit convergence as the displacement decreases. In contrast, Figures Figure C.11 and Figure C.12 display the phonon bands for the same three materials predicted using eqV2-S-DeNS (direct-forces), which not only fail to demonstrate convergence but also exhibit significant errors, including missing

---

[2]In addition to our model, we also run the evaluation for GRACE (Bochkarev et al., 2024), SevenNet (Park et al., 2024), Orb (Neumann et al., 2024), and eqV2-S-DeNS (Liao et al., 2023; Barroso-Luque et al., 2024), which were not included in the work by Loew et al. 2024.

*Table 4.* Summary of model performance on the MDR Phonon benchmark. Metrics include maximum phonon frequency (MAE($\omega_{max}$), in K), the vibrational entropy (MAE($S$), in J/K/mol), the Helmholtz free energy (MAE($F$), in kJ/mol), and the heat capacity at constant volume (MAE($C_V$), in J/K/mol). *The OMat24 dataset uses a slightly different DFT setting from the DFT setting of the MDR Phonon benchmark.

| Compliant models | MAE($\omega_{max}$) | MAE($S$) | MAE($F$) | MAE($C_V$) |
|---|---|---|---|---|
| M3GNet | 98 | 150 | 56 | 22 |
| CHGNet | 89 | 114 | 45 | 21 |
| MACE | 61 | 60 | 24 | 13 |
| GRACE-2L (r6) | 40 | 25 | 9 | 5 |
| SevenNet-0 | 40 | 48 | 19 | 9 |
| SevenNet-l3i5 | 26 | 28 | 10 | 5 |
| eSEN-30M-MP | **21** | **13** | **5** | **4** |
| *Direct-force models* | | | | |
| Orb MPTrj [0.01 Å] | 309 | 476 | 64 | 181 |
| Orb MPTrj [0.2 Å] | 61 | 34 | 11 | 8 |
| eqV2-S-DeNS [0.01 Å] | 280 | 224 | 54 | 94 |
| eqV2-S-DeNS [0.2 Å] | 58 | 26 | 8 | 8 |
| *Non-compliant models* | | | | |
| eqV2-M-OAM [0.01 Å] | 780 | 403 | 241 | 100 |
| eqV2-M-OAM [0.2 Å] | 50 | 25 | 7 | 9 |
| MatterSim | 17 | 15 | 5 | 3 |
| GRACE-2L-OAM | 19 | 14 | 5 | 4 |
| SevenNet-MF-ompa | **15** | **8** | **3** | **3** |
| eSEN-30M-OAM | **15** | 10 | 4 | **3** |

acoustic branches and spurious imaginary frequencies.

While we find that the OMat-trained models (without sAlex/MPTrj finetuning) may provide a lower error on phonon prediction (7/7/2/2 for $\omega_{max}/S/F/C_V$ MAE), we refrain from direct comparison due to mismatch in level of theory. We attribute this result to the softening issue of the sAlex and MPTrj dataset (Deng et al., 2025; Barroso-Luque et al., 2024), which OMat24 addresses. We refer interested

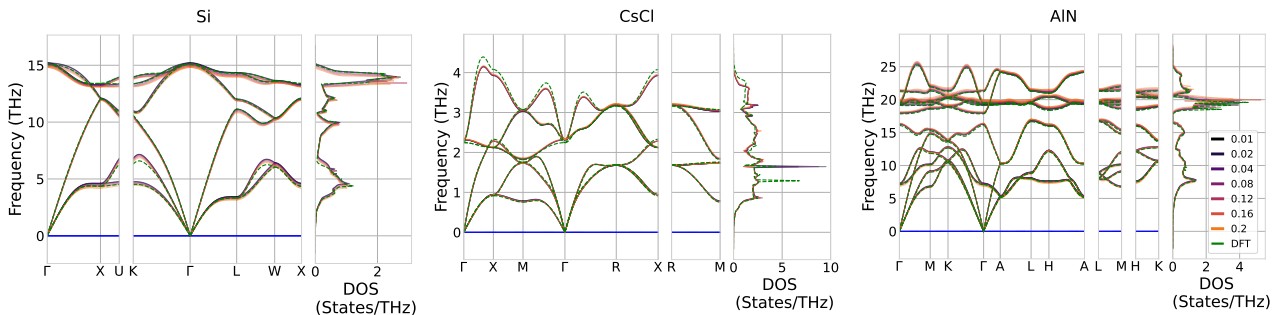

*Figure 5.* Predicted phonon band structure and density of states (DOS) of Si (diamond structure), CsCl (CsCl structure), AlN (wurtzite structure) using eSEN at different displacement values. DFT baseline is taken from the PBE MDR dataset (Loew et al., 2024) calculated using a displacement of 0.01 Å.

*Table 5.* Test set MAE for SPICE-MACE-OFF. Energy (**E**) MAE is in meV/atom. Force (**F**) MAE is in meV/Å. [*]EscAIP-45M is a direct-force model.

| | MACE-4.7M | | EScAIP-45M[*] | | eSEN-3.2M | | eSEN-6.5M | |
|---|---|---|---|---|---|---|---|---|
| Dataset | **E** | **F** | **E** | **F** | **E** | **F** | **E** | **F** |
| PubChem | 0.88 | 14.75 | 0.53 | 5.86 | 0.22 | 6.10 | **0.15** | **4.21** |
| DES370K M. | 0.59 | 6.58 | 0.41 | 3.48 | 0.17 | 1.85 | **0.13** | **1.24** |
| DES370K D. | 0.54 | 6.62 | 0.38 | 2.18 | 0.20 | 2.77 | **0.15** | **2.12** |
| Dipeptides | 0.42 | 10.19 | 0.31 | 5.21 | 0.10 | 3.04 | **0.07** | **2.00** |
| Sol. AA | 0.98 | 19.43 | 0.61 | 11.52 | 0.30 | 5.76 | **0.25** | **3.68** |
| Water | 0.83 | 13.57 | 0.72 | 10.31 | 0.24 | 3.88 | **0.15** | **2.50** |
| QMugs | 0.45 | 16.93 | 0.41 | 8.74 | 0.16 | 5.70 | **0.12** | **3.78** |

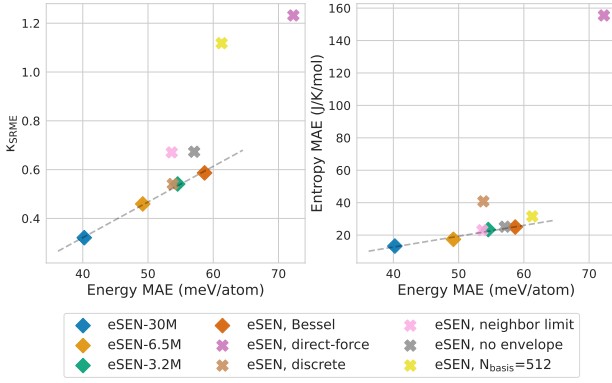

readers to Deng et al. 2025 and Barroso-Luque et al. 2024 for a detailed discussion on the softening issue of some DFT datasets and its relation to phonon properties.

### 6.3. SPICE-MACE-OFF

We train and evaluate eSEN models on the SPICE-MACE-OFF dataset (Kovács et al., 2023), which is built upon the SPICE dataset (Eastman et al., 2023). As shown in Table 5, eSEN with 6.5M parameters outperforms MACE-OFF-L (4.7M parameters) and EscAIP (45M parameters, direct-force) on all test-set splits for both energy and force MAE. We also include results for eSEN with 3.2M parameters, which has inference efficiency similar to MACE-4.7M, while achieving lower test energy/force MAE. More details on the inference efficiency benchmark are included in Appendix C.

### 6.4. Test-set error for model development

In Figure 1, we showed the correlation between test error and physical property prediction tasks for different architectures. Figure 6 demonstrates this correlation for different variants of eSEN (with a 1k-materials subset of the MDR Phonon benchmark for efficiency). In particular, among models that pass the MD energy conservation test, a strong

*Figure 6.* Test error correlation across several property prediction tasks for eSEN variants. Conservative models are shown as boxes and those found to not conserve as crosses. Note metrics for conservative models have a stronger correlation with test set errors.

correlation between test error and $\kappa_{\mathrm{SRME}}$/vibrational entropy MAE can be observed. We include experimental details about Figure 1 and Figure 6 in Appendix A and additional results for other phonon properties in Appendix B.

## 7. Related works

**MLIP architectures** have made significant progress since their initial proposal (Behler & Parrinello, 2007). These architectures are usually symmetry-preserving (Smith et al., 2017; Schütt et al., 2017; Gilmer et al., 2017; Chmiela et al., 2017; Artrith et al., 2017; Unke & Meuwly, 2018; Zhang et al., 2018; Zubatyuk et al., 2019; Smith et al., 2020; Kovács et al., 2021), with increasingly expressive atom environment embeddings and message-passing operations (Gasteiger et al., 2020b; 2021; Schütt et al., 2021; Liu et al., 2021; Unke et al., 2021a; Chen & Ong, 2022; Deng et al., 2023; Cheng, 2024; Yin et al., 2025). Notably, equiv-

ariant architectures based on spherical harmonics representations (Thomas et al., 2018; Thölke & De Fabritiis, 2021; Batzner et al., 2022; Musaelian et al., 2022; Batatia et al., 2022; Passaro & Zitnick, 2023; Liao et al., 2023; Bochkarev et al., 2024; Park et al., 2024; Batatia et al., 2025) have shown strong performance on large-scale datasets. Meanwhile, the high computational cost of these architectures has sparked significant interest in scalable architectures that may not respect physical principles such as energy conservation (Langer et al., 2024; Brehmer et al., 2024; Hu et al., 2021; Yang et al., 2024; Qu & Krishnapriyan, 2024; Neumann et al., 2024; Rhodes et al., 2025). These models have demonstrated strong performance in accuracy, scalability, and relaxation tasks (Chanussot et al., 2021; Riebesell et al., 2023). While their non-physical nature may make them unsuitable for direct usage in some physical property prediction tasks, they may still provide benefit by using the pre-training strategy proposed in this paper and Bigi et al. 2024, distilling them to conservative models (Amin et al., 2025), or combining them with a conservative model using multiple-time-step integration (Bigi et al., 2024).

**MLIPs and physical observables.** While MLIPs continue to improve, it is necessary to evaluate them in realistic tasks that are relevant to scientific discovery. Physical property prediction benchmarks that involve geometry optimization (Riebesell et al., 2023; Lan et al., 2023; Wander et al., 2024), MD simulations (Fu et al., 2023; Kovács et al., 2023; Moore et al., 2024; Sabanes Zariquiey et al., 2024; Eastman et al., 2024), vibrational analysis and phonon calculations (Póta et al., 2024; Loew et al., 2024; Wines & Choudhary, 2024), and others are increasing in scale with broader applications and wider adoption. Training strategies for learning from physical observables (Wang et al., 2020; Greener, 2024; Röcken et al., 2024; Raja et al., 2024) and the higher-order derivatives of the PES (Fang et al., 2024; Williams et al., 2025) are promising directions to further improve MLIPs for predicting physical properties.

## 8. Discussion

We identify conservative forces and a smoothly-varying PES as two important properties for MLIPs to consistently perform well in physical property prediction tasks. We offer an analysis of design choices to enhance these two properties. The resulting eSEN architecture bridges the gap between the test-set error and downstream applications, achieving SOTA performance in force/energy prediction, geometry optimization, phonon calculations, and thermal conductivity prediction. This implies it may be possible to use test error as a proxy metric for evaluating model performance during development, if a model passes energy conservation tests. This can accelerate innovations in MLIPs, since benchmarking physical properties usually requires significant domain

knowledge and is usually time-consuming, whereas evaluating test set error is straightforward and efficient.

## Software and Data

Code is available at: https://github.com/facebookresearch/fairchem. Pretrained checkpoints are available at https://huggingface.co/facebook/OMAT24.

## Acknowledgements

We acknowledge Zachary W. Ulissi (FAIR at Meta), Aditi S. Krishnapriyan (UC Berkeley), Samuel M. Blau (LBNL) and other members of the FAIR Chemistry team for helpful discussions, and Ammar Rizvi (FAIR at Meta) for project support.

## Impact Statement

This paper presents work whose goal is to advance the field of machine learning for atomistic modeling, with the potential to accelerate scientific discovery in chemistry and biological sciences. As the field of ML-based atomistic modeling continues to evolve, we emphasize the importance of ongoing discussions on ethical considerations and best practices to ensure that these technologies are developed and applied in ways that benefit society as a whole.

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

# A. Experimental details

## A.1. MD simulation protocol

Simulating molecular systems not seen during training is a key capability of MLIPs. Therefore, we construct challenging MD simulation tasks featuring out-of-distribution data to test a model's conservation capability. We conduct experiments on two chemical domains: inorganic crystals and organic molecules.

**Inorganic materials.** For training models on inorganic materials we utilize the MPTrj (Deng et al., 2023) dataset, and we establish a suite of simulation tasks based on the TM23 dataset (Owen et al., 2024). TM23 contains MD samples of 27 single vacancy defect transition metal systems at cold, warm, and melt temperatures using an NVT ensemble, in total 81 combinations of different metals and temperature. These defected systems are out-of-distribution for a model trained on the MPTrj datasets, which only contains relaxation trajectories of non-defected systems. Additionally, some of the metals in the TM23 dataset is very rare in the MPTrj dataset. We initialize the simulation by sampling a frame from the TM23 dataset, run a relaxation using the LBFGS algorithm, randomly initialize the atom velocities at cold/warm/melt temperatures using a Maxwell-Boltzmann distribution, then run MD simulations under the NVE ensemble for 100 ps using a time step of 5 fs (same as the time step used in the TM23 ab initio MD protocol).

**Organic molecules.** For training models on organic molecules we use the SPICE-MACE-OFF dataset (Kovács et al., 2023), which is mainly based on the SPICE-1.0 dataset (Eastman et al., 2023), and we establish a suite of simulation tasks from the MD22 dataset (Chmiela et al., 2023) that contains seven large molecules. Molecules in MD22 are out-of-distribution for a model trained on the SPICE-MACE-OFF dataset as they are considerably larger than all molecules in the SPICE-MACE-OFF training dataset. We initialize the simulation by sampling a frame from the MD22 dataset, run a relaxation using the LBFGS algorithm, randomly initialize the atom velocities at a temperatures of 400/500 K using a Maxwell-Boltzmann distribution (400 K for Buckyball catcher and Double-walled nanotube and 500 K for other molecules, which are the same as the MD22 protocol), then run MD simulations under the NVE ensemble for 100 ps using a time step of 1 fs (same as the time step used in the MD22 ab initio MD protocol).

All ML-based MD simulations use a Velocity-Verlet integrator and are conducted with ASE (Larsen et al., 2017). We measure the energy conservation error (extent of energy drift) across the 100-ps simulations. All eSEN models are 2-layer with $L_{max} = 2$ and $M_{max} = 2$ (3.2M trainable parameters). Detailed model hyperparameters are included in Appendix D.

## A.2. Phonon calculation protocols

Harmonic and anharmonic phonon calculations and solutions to the Wigner transport equation (Simoncelli et al., 2022) used for the thermal conductivity benchmark ($\kappa_{SRME}$) values given in Table 2 were carried out using the supercell method with finite differences implemented in PHONO3PY (Togo et al., 2015; 2023). The calculations followed the protocol described in the Matbench-Discovery benchmark (Riebesell et al., 2023; Póta et al., 2024). For eSEN-30M, an even lower $\kappa_{SRME}$ of 0.298 is obtained when we adjust the evaluation parameter atom displacement from 0.03 Å to 0.05 Å.

Harmonic phonon calculations for the MDR benchmark results listed in Section 6.2 were carried out following the calculation protocol used by (Loew et al., 2024) which employs phonon calculations using the supercell method with finite differences with a displacement of 0.01 Å. Calculations were done using the PHONOPY software (Togo et al., 2023).

## A.3. Test-set error for MPTrj-trained models

Since MPTrj lacks an official test split and various models are typically trained on distinct subsets of the data, we randomly selected 5000 samples from the subsampled Alexandria (sAlex) dataset (Schmidt et al., 2024; Barroso-Luque et al., 2024) for a fair comparison. This subset was used to calculate the test-set energy mean absolute errors (MAEs) presented in Figures 1, 6, B.7, and B.8.

In the ablation study reported in Table 1, we use a random test split of the MPTrj dataset. We find the performance of eSEN to be highly stable with different random seeds. We trained 2-layer eSENmodels on the MPTrj dataset using 3 different seeds for 50 epochs, with loss coefficients the same loss coefficients of models in Table 1. The validation set errors and standard deviations are $19.67 \pm 0.23$ for energy (meV/atom), $43.85 \pm 0.058$ for forces (meV/Å), and $0.16 \pm 0.00038$ for stress (meV/(Å$^3$· atom)).

# B. Phonon calculations and vibrational properties

## B.1. Correlation of test-set energy errors and vibrational property errors

Figure B.7 presents the correlation between test-set energy MAE and the other three phonon calculation tasks, evaluated across various model architectures. The corresponding correlations for different variants of eSEN are displayed in Figure B.8. In both figures, improved correlation can be observed among energy-conserving models.

Figure B.8 illustrates that failing the conservation test can

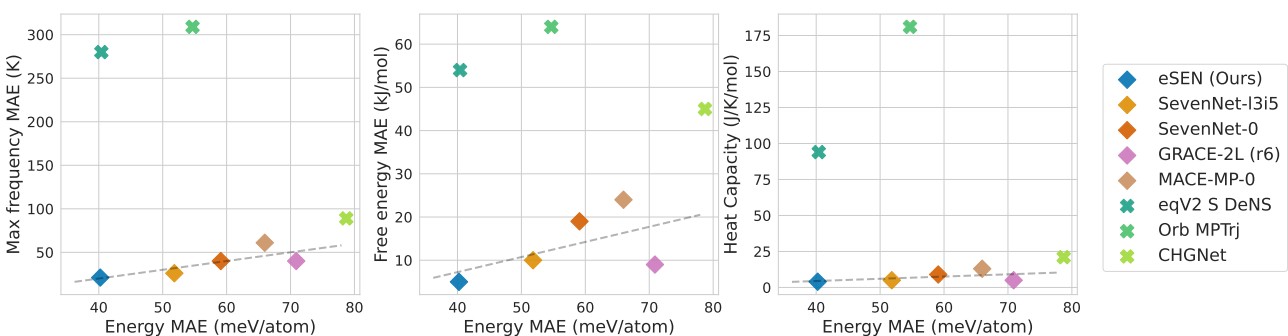

*Figure B.7.* The correlation between test-set energy error and maximum frequency, free energy and heat capacity across different model architectures.

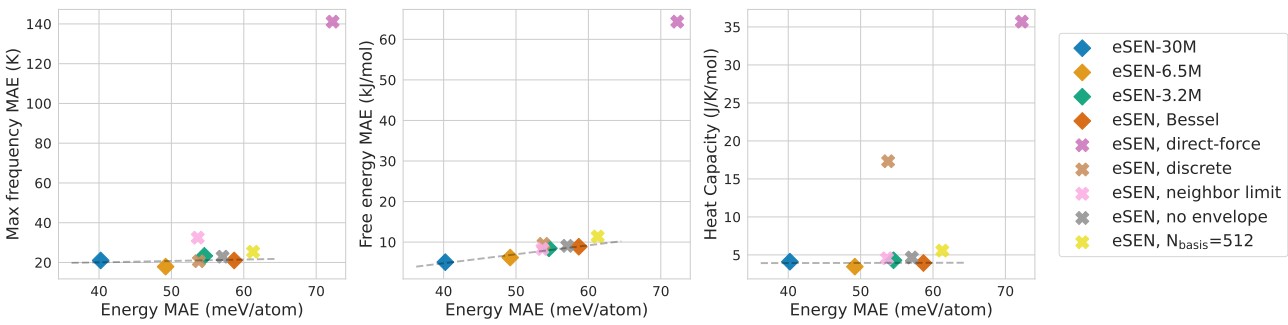

*Figure B.8.* The correlation between test-set energy error and maximum frequency, free energy and heat capacity for different variants of eSEN.

result in varying degrees of impact on different properties, depending on the specific design choices made. Although the neighbor limit, envelope function, and number of basis functions substantially affect $\kappa_{\mathrm{SRME}}$ (see Figure 6), their influence on the properties evaluated in the MDR Phonon benchmark is relatively minor. Representation discretization impacts vibrational entropy and heat capacity but not other properties. While a model might still be able to get good performance in some physical property task when the energy conservation test is failed, when the conservation test is passed, the model performs very well robustly across all metrics.

### B.2. Displacement values and their relation to phonon band structure predictions

Figure B.9 presents the MAEs on phonon calculations for eSEN, MACE, and eqV2 S DeNS as a function of increasing displacement values. As anticipated, both eSEN and MACE exhibit constant or slightly increasing MAE with respect to displacement. In contrast, eqV2 S DeNS displays a notable decrease in MAE with increasing displacement. Notably, when using a displacement of 0.2 Å, the resulting phonon

benchmark MAE values for eqV2 S DeNS become comparable to those of conservative force models (Section 6.2).

While the prediction accuracy of thermodynamic properties such as maximum frequency, entropy, free energy, and heat capacity improves with increasing displacement for direct-force models, this improvement is deceptive and does not translate to accurate predictions of the underlying phonon band structure and density of states (DOS). As illustrated in Figure C.11, the predicted phonon bands and DOS for three selected materials exhibit significant errors, particularly in capturing the correct dispersion relations. Moreover, imaginary frequencies are commonly predicted at small displacement values, suggesting a rough energy landscape (i.e. the learned PES is not truly convex when it's very close to the minima). The eqV2 S DeNS model also fails to accurately capture acoustic modes—those that go to zero linearly at the $\gamma$ point—which is due to a non-zero net force on the structure. In contrast, non-zero net force at energy local minima does not occur for conservative models by definition (Figure 5).

By enforcing a net zero force prediction, as proposed by Neumann et al. 2024, direct-force models can be modified

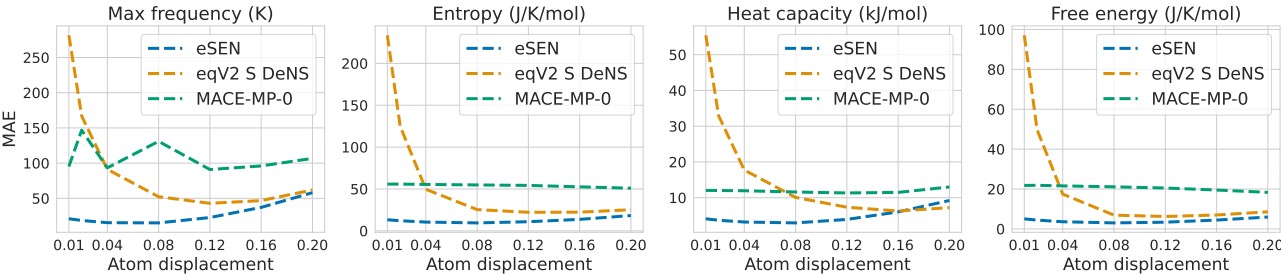

*Figure B.9.* Errors in a randomly sampled subset (1000 samples) of the MDR Phonon benchmark when the atom displacement is adjusted.

to accurately capture acoustic phonon modes. As demonstrated in Figure C.12, incorporating this constraint allows the model to predict acoustic modes correctly. However, despite this improvement, the model continues to struggle with accurately reproducing the phonon band structure, and the issue of predicting imaginary frequencies at small displacement values persists.

The apparent paradox of direct-force models like eqV2 S DeNS failing to accurately capture phonon band structures while still achieving competitive accuracy for thermodynamic properties such as entropy, free energy, and heat capacity can be resolved by examining the underlying calculation methodology. These properties are computed using weighted integrals of the DOS. Additionally the metrics in the MDR are performed at room temperature (300 K). The predicted DOS in Figure C.11 and Figure C.12 at larger displacement values adequately captures the overall features of the DFT DOS, but does not reproduce finer details such as high density areas and fluctuations. This level of agreement is sufficient for accurate predictions because the Boltzmann-weighted integrals used in calculating thermal properties help to mitigate the impact of point-wise errors, making it less crucial to precisely capture fine details in the band structure and DOS (Ackland et al., 1997; Van De Walle & Ceder, 2002). Moreover, since properties are estimated at 300 K, models can achieve accurate predictions by prioritizing prediction accuracy of lower-frequency modes, which are more relevant for thermal property calculations, rather than attempting to capture higher-frequency modes. Although eqV2 S DeNS without a net-zero force contraint may not accurately capture acoustic phonon branches, its ability to predict vibrational thermodynamics at room temperature remains unaffected due to the relatively small number of acoustic phonon states at low frequencies.

As a comparison with eqV2 S DeNS, Figures Figure C.13 and Figure C.14 shows the predicted phonon dispersion and DOS for the same three materials using eSEN with direct-force prediction. Although the results still exhibit some of the characteristic artifacts of direct-force models, such

as convergence at larger displacements and the absence of acoustic modes, these issues are less pronounced compared to eqV2 S DeNS. Moreover, the predicted phonon bands and DOS are significantly improved, providing a more accurate representation of the DFT reference values. The better approximation of phonon bands and a lower tendency to predict imaginary frequencies highlight the importance of a smoothly-varying model, even without being conservative.

Extending the existing metrics proposed by Loew et al. 2024 with additional evaluations would provide a more comprehensive assessment of MLIP performance. Specifically, new metrics could be developed to assess phonon dispersion across all modes and frequencies at commensurate points; and computing vibrational thermodynamic properties at a range of temperatures.

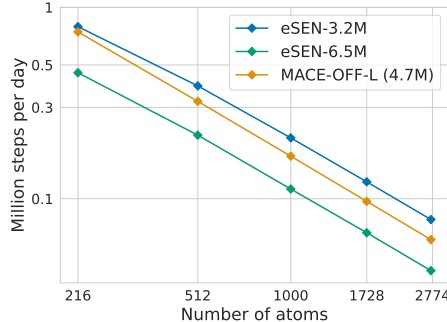

*Figure B.10.* Inference efficiency of MACE-OFF-L and eSENs of a similar scale.

## C. Inference efficiency

We benchmarked the inference speed of our models against the similar sized MACE-OFF-L (Kovács et al., 2023) (4.7M) on a single 80GB Nvidia A-100 GPU. For MACE-OFF-L we used the exact benchmark code found in `https://github.com/ACEsuit/mace/blob/main/tests/test_benchmark.py` with mace-torch v0.3.6 (PyPi). To create a fair comparison, we replicated the identical benchmark environment as MACE benchmarks

using the same diamond system with variable number of supercells (carbon atoms) as input. All models are benchmarked using the standard Python (3.12) runtime with Pytorch v2.4.0 (Paszke et al., 2019) and CUDA 12.1 (Nickolls et al., 2008). No compile/torchscript was used for standardization of runtime. Across all system sizes, eSEN-3.2M has a comparable inference efficiency to MACE-OFF-L. For 216 atoms (Figure B.10), our models (3.2M, 6.4M) can run approximately (0.4, 0.8) million steps per day comparable to MACE-OFF-L (0.7 million steps per day).

## D. Hyper-parameters

Hyper-parameters used for model training are shown in Table 6 We train all models using a per-atom energy MAE loss, a force $l_2$ loss, and a stress MAE loss. For direct-force models or direct-force pre-training, we use the same decomposed loss as described in Barroso-Luque et al. 2024. The eSEN-30M-MP model trained on MPTrj uses Denoising Non-equilibrium Structures (DeNS) (Liao et al., 2024) with a noising probablity of 0.5, a standard deviation of 0.1 Å for the added Gaussian noise, and 10 for the DeNS loss coefficient during direct-force pre-training. DeNS is not used during conservative fine-tuning. In our ablation study for maximum neighbor limit we used 30 as the limit. The main effect of DeNS is reducing overfitting in the pretraining stage. Such benefit is found to carry over to the conservative finetuning stage. The eSEN-30M-OMat and the OAM Fine-tuned models do not use DeNS.

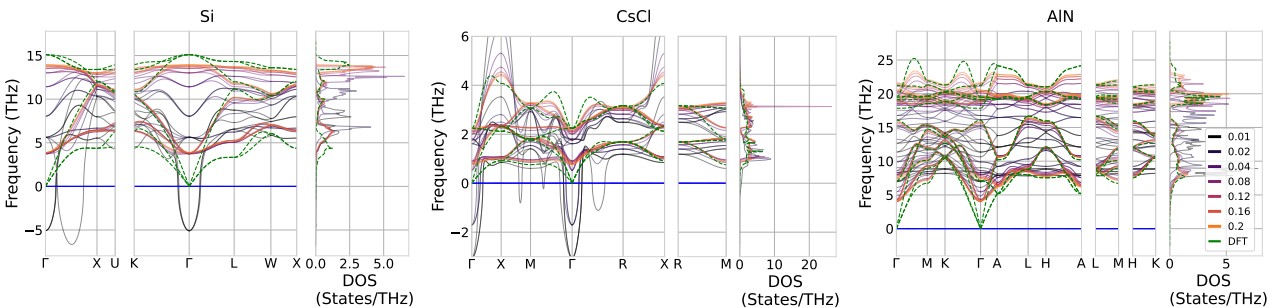

*Figure C.11.* Predicted phonon band structure and density of states (DOS) of Si (diamond structure), CsCl (CsCl structure), AlN (wurtzite structure) using eqV2 S DeNS (direct-force prediction) at different displacement values. DFT baseline is taken from the PBE MDR dataset (Loew et al., 2024) calculated using a displacement of 0.01 Å

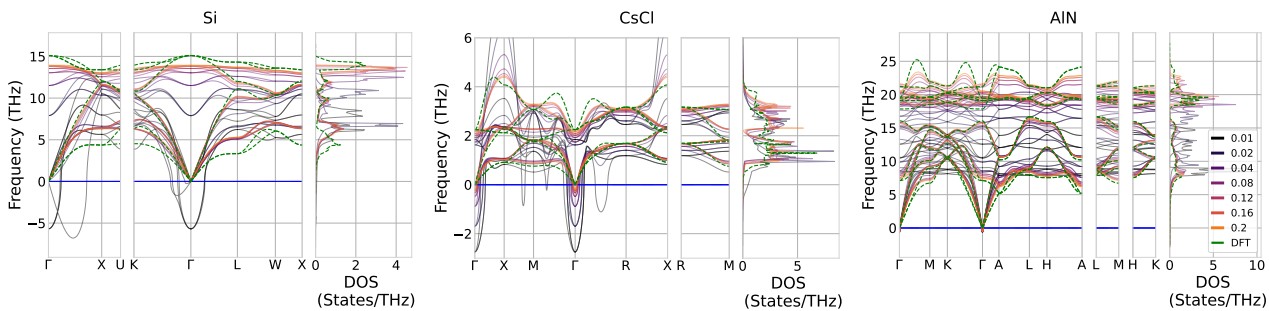

*Figure C.12.* Predicted phonon band structure and density of states (DOS) of Si (diamond structure), CsCl (CsCl structure), AlN (wurtzite structure) using eqV2 S DeNS (direct-force prediction with a zero net force constraint) at different displacement values. DFT baseline is taken from the PBE MDR dataset (Loew et al., 2024) calculated using a displacement of 0.01 Å

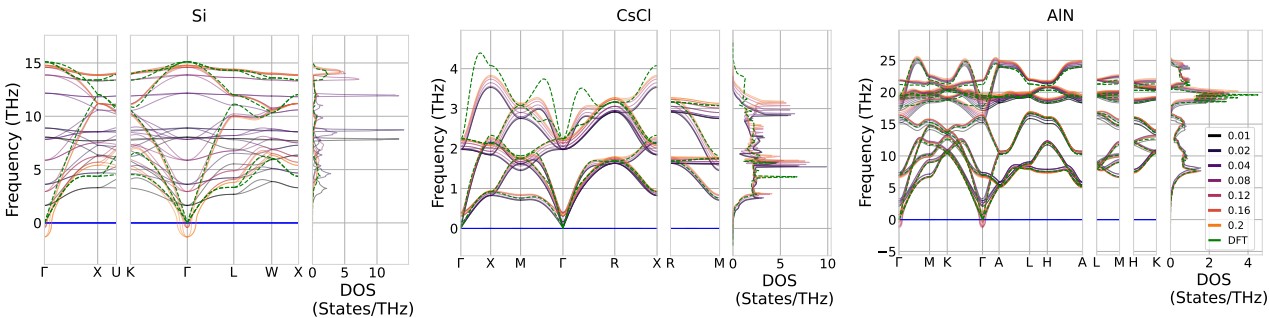

*Figure C.13.* Predicted phonon band structure and density of states (DOS) of Si (diamond structure), CsCl (CsCl structure), AlN (wurtzite structure) using eSEN with direct-force prediction at different displacement values. DFT baseline is taken from the PBE MDR dataset (Loew et al., 2024) calculated using a displacement of 0.01 Å

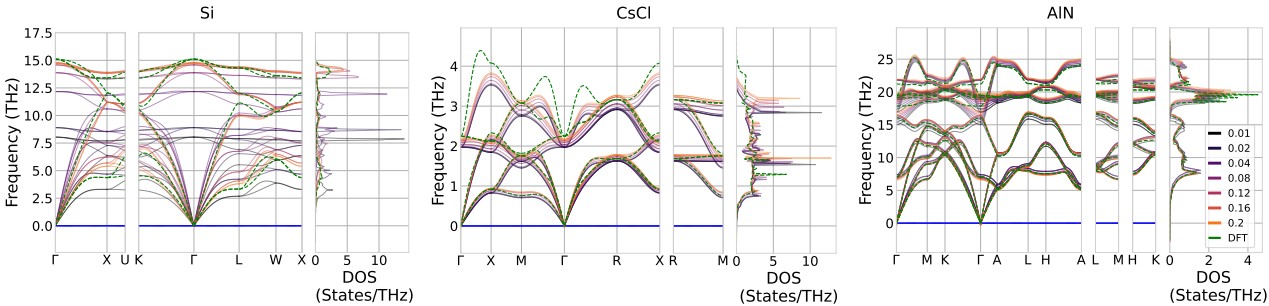

*Figure C.14.* Predicted phonon band structure and density of states (DOS) of Si (diamond structure), CsCl (CsCl structure), AlN (wurtzite structure) using eSEN with direct-force prediction and a zero-net force constraint at different displacement values. DFT baseline is taken from the PBE MDR dataset (Loew et al., 2024) calculated using a displacement of 0.01 Å

*Table 6.* Hyper-parameters for eSEN variants reported in this paper. *eSEN-30M on MPTrj was trained for 60 epochs using direct-force pre-training and 40 epochs of conserved fine-tuning. eSEN-30M-OMat was trained for 2 epochs using direct-force pre-training and 2 epochs of conserved fine-tuning. †The eSEN-30M-OAM model starts from the eSEN-30M-OMat model, and was finetuned for 1 epoch on a dataset constructed by combining the sAlex training dataset and 8 copies of the MPTrj training dataset.

| Hyper-parameters | SPICE-3.2M | SPICE-6.5M | MPTrj-3.2M | MPTrj-6.5M | MPTrj-30M | OMat-30M | OAM Fine-tuning |
|---|---|---|---|---|---|---|---|
| Number of eSEN layer blocks | 2 | 4 | 2 | 4 | 10 | 10 | 10 |
| Maximum degree $L_{max}$ | 2 | 2 | 2 | 2 | 3 | 3 | 3 |
| Maximum order $M_{max}$ | 2 | 2 | 2 | 2 | 2 | 2 | 2 |
| Number of channels $N_{channel}$ | 128 | 128 | 128 | 128 | 128 | 128 | 128 |
| Radial basis function | Bessel | Bessel | Gaussian | Gaussian | Gaussian | Gaussian | Gaussian |
| Number of radial basis functions | 10 | 10 | 10 | 10 | 10 | 64 | 64 |
| Cutoff radius (Å) | 5 | 5 | 6 | 6 | 6 | 6 | 6 |
| Batch size | 128 | 128 | 512 | 512 | 512 | 512 | 256 |
| Optimizer | AdamW | AdamW | AdamW | AdamW | AdamW | AdamW | AdamW |
| Learning rate scheduling | Cosine | Cosine | Cosine | Cosine | Cosine | Cosine | Cosine |
| Warmup epochs | 0.1 | 0.01 | 0.1 | 0.1 | 0.1 | 0.1 | 0.1 |
| Warmup factor | 0.2 | 0.2 | 0.2 | 0.2 | 0.2 | 0.2 | 0.2 |
| Maximum learning rate | $4 \times 10^{-4}$ | $4 \times 10^{-4}$ | $4 \times 10^{-4}$ | $4 \times 10^{-4}$ | $4 \times 10^{-4}$ | $4 \times 10^{-4}$ | $2 \times 10^{-4}$ |
| Number of epochs | 100 | 100 | 100 | 100 | $60 + 40^*$ | $2 + 2^*$ | $1^†$ |
| Gradient clipping norm | 100 | 100 | 100 | 100 | 100 | 100 | 100 |
| Model EMA decay | 0.999 | 0.999 | 0.999 | 0.999 | 0.999 | 0.999 | 0.999 |
| Weight decay | $1 \times 10^{-3}$ | $1 \times 10^{-3}$ | $1 \times 10^{-3}$ | $1 \times 10^{-3}$ | $1 \times 10^{-3}$ | $1 \times 10^{-3}$ | $1 \times 10^{-3}$ |
| Energy loss coefficient | 10 | 10 | 1 | 1 | 20 | 20 | 20 |
| Force loss coefficient | 20 | 20 | 10 | 10 | 20 | 20 | 20 |
| Stress loss coefficient | - | - | 100 | 100 | 5 | 5 | 5 |

