# OpenReview forum: "Learning Smooth and Expressive Interatomic Potentials for Physical Property Prediction"
_ICML.cc/2025/Conference — ICML 2025 oral_

### Official Review · Reviewer_yR9i · 2025-03-12

**Overall Recommendation:** 5

**Summary:**

This paper argues that the test MAE, when energy conservation is guaranteed in MD simulations, demonstrates the practicality of machine learning potentials. The authors provide empirical evidence indicating that, within established model designs, specific designs uphold energy conservation principles while others do not. Furthermore, they demonstrate a correlation between the Energy/Force MAE on the test dataset and the predictive performance of physical properties utilizing MD simulations, specifically for those models that maintain energy conservation. The resulting model, eSEN, achieves state-of-the-art results across various physical property prediction tasks based on phonon calculations.

## update after rebuttal
Thank you for the author's responses. This paper is valuable to share with the community, and I support its publication.

**Claims And Evidence:**

It has been experimentally demonstrated that the energy conservation law, which is a property that the potential must satisfy, affects the estimation accuracy of physical quantities that require phonon calculations.
The model design, which is well-known and of interest to readers, has been experimentally verified and supported by evidence. However, it should be noted somewhere in the text that the physical properties considered in this study are limited to those requiring higher-order derivatives of the PES and that the applications of machine learning potentials have not been examined comprehensively.

**Essential References Not Discussed:**

I do not find any issue with the references.

**Experimental Designs Or Analyses:**

I have checked the NVE MD simulations to verify the energy conservation law, as well as the experimental settings for Matbench Discovery and the MDR phonon benchmark. There are no issues with these experimental settings.

**Methods And Evaluation Criteria:**

It is reasonable to evaluate the performance of a potential that satisfies the energy conservation law using physical properties that require differentiation.

**Other Comments Or Suggestions:**

I am wondering about the contribution of DeNS to thermodynamics property predictions, and I would like to see DeNS's ablation study.

**Other Strengths And Weaknesses:**

Strengths
- Conservative fine-tuning could be a popular method in the community since it speeds up the training without sacrificing energy conservation.

**Questions For Authors:**

No question.

**Relation To Broader Scientific Literature:**

The machine learning community has proposed designs that differ from conventional potential designs. However, some of these machine learning potentials, despite having good MAE for energy and forces, are not practical for real-world use, and it has been repeatedly pointed out that Energy/Force MAE does not necessarily indicate the practicality of machine learning potentials.
Issues with the practicality of these potentials can be observed during actual simulations, such as structural failures, but a simple and broadly applicable method to identify these problems was not previously known.

**Theoretical Claims:**

The results are empirical, so there is no theoretical claim.

Additionally, I have briefly reviewed Hairer et al. 2003 and believe that it aligns with the claims in Section 3.2. However, in my understanding, Section 3.2 does not present any novel theoretical claims.

---

> ### Author Rebuttal · Authors · 2025-03-27
>
> We thank reviewer yR9i for the helpful feedback. We address each of the reviewer’s comments below.
>
> > It should be noted somewhere in the text that the physical properties considered in this study are limited to those requiring higher-order derivatives of the PES and that the applications of machine learning potentials have not been examined comprehensively.
>
> Thank you for your suggestion. While other properties such as formation energy in the Matbench-Discovery benchmark were studied in the paper, the properties that require higher-order derivatives of the PES are indeed more significantly impacted by the MD-energy-conservation properties. We will note this in the revised manuscript.
>
> > I am wondering about the contribution of DeNS to thermodynamics property predictions, and I would like to see DeNS's ablation study.
>
> Only conservative models are well-suited for thermodynamics property prediction tasks, which require accurate modeling of higher-order PES derivatives. DeNS is only used during direct pre-training on the MPTrj dataset to alleviate overfitting. Conservative models do not use DeNS during conservative training. We present an ablation study over two 2-layer direct-force eSEN models (with loss coefficients E: 1/F: 10/S: 100), with and without DeNS:
>
> | Metric                          | With DeNS | Without DeNS |
> |-----------------------------------|-----------|--------------|
> | Energy MAE (meV/atom)         | 18.0      | 19.4         |
> | Force MAE (meV/Å)             | 43.7      | 43.7         |
> | Stress MAE (meV/Å³ atom)      | 0.14      | 0.16         |
>
> The higher error of the model without DeNS is due to overfitting. We will incorporate the validation error curves in the revised manuscript to reflect that.
>
> We look forward to further discussions if you have additional questions or suggestions. Thank you again for your valuable input.

---

> > ### Comment · Reviewer_yR9i · 2025-04-03
> >
> > Thank you for your response.
> > Is the result you provided a comparison of the accuracy after performing conservative training following direct-force pretraining with/without DeNS? Can it be said that using DeNS during pretraining has influenced the accuracy of the model after conservative training?

---

> > > ### Author Response · Authors · 2025-04-05
> > >
> > > Thank you for the additional comment! This is an excellent point -- the results above are from direct-force models without conservative training. We have started running new experiments with conservative training and will update the results here once they finish.
> > >
> > > Update:
> > >
> > > we pretrain direct-force eSEN models (2-layer, loss coefficients E: 1/F: 10/S: 100) for 60 epochs with and without DeNS, followed 40 epochs of conserved training without DeNS. The validation errors are:
> > >
> > > | Property                          | With DeNS | Without DeNS |
> > > |-----------------------------------|-----------|--------------|
> > > | Val Energy MAE (meV/atom)         | 17.6      | 19.3         |
> > > | Val Force MAE (meV/Å)             | 43.1      | 44.0         |
> > > | Val Stress MAE (meV/Å³ atom)      | 0.14      | 0.14         |
> > >
> > > We find the effect of DeNS at the direct-force training stage carries over to the final conservative models.

---

### Official Review · Reviewer_8LdR · 2025-03-13

**Overall Recommendation:** 5

**Summary:**

This paper draws attention to the inability of energy conservation, and thereby instability of simulation, common in many popular machine learning interatomic potentials (MLIPs). Next, it proposes a novel architecture addressign this problem, while showing state-of-the-art performance on a wide range of tasks.

**Claims And Evidence:**

Yes. The claims were all well supported experimentally. The discussion around the force conservation is particularly exhilarating and well supported.

**Essential References Not Discussed:**

NA.

**Experimental Designs Or Analyses:**

The experiments are designed properly and convincingly. In particular, they refreshingly used larger, more realistic datasets rather than the toy datasets the field has been using.

**Methods And Evaluation Criteria:**

The proposed methods and evaluation criteria are very reasonable. Apart from the conservation errors, the ensemble property experiments proposed in this paper also throw new light on the discussion around the desired properties of MLIP.

**Other Comments Or Suggestions:**

NA.

**Other Strengths And Weaknesses:**

NA.

**Questions For Authors:**

Can the claims made in Section 5 be supported by theoretical arguments?

**Relation To Broader Scientific Literature:**

The relevant papers have been properly cited and discussed.

**Theoretical Claims:**

NA. There is little theoretical claims.

---

> ### Author Rebuttal · Authors · 2025-03-27
>
> We thank reviewer 8LdR for the helpful feedback. We address each of the reviewer’s comments below.
>
> > Can the claims made in Section 5 be supported by theoretical arguments?
>
> We refer to (1) Hairer et al. 2003 for theoretical arguments on the relationship between potential energy surface (PES) smoothness/bounds on derivatives and energy conservation in simulations; and (2) Molecular simulation textbooks (e.g., Tuckerman, 2023) for theoretical arguments on conservative forces. Section 5 is then organized to discuss practical implementations of machine learning potentials that impact conservative forces, smoothness and bounds on derivatives. Some arguments on the design choices can be made from a theoretical perspective:
>
> - **Direct-force:** a direct-force model the direct-force prediction framework imposes no constraint on the output forces being a conservative force field. Using the derivative of the energy with respect to the atomic positions is a requirement for ensuring the predicted forces are conservative.
> - **Representation discretization:** The discretization of spherical harmonics representations breaks the conservative forces requirement because it introduces discretization errors to the computation of energy gradients. Increasing the grid resolution theoretically reduces such discretization errors and helps conservation.
> - **Max neighbor limit:** From a theoretical perspective, we can show examples where the K-NN graph introduces PES discontinuity. Consider a model with a cutoff of 6 A, and a node having 3 neighbors at distance 3 A and a fourth neighbor at (3 + epsilon) A. a small perturbation to the atomic positions will introduce discontinue change in the predicted energy, if a max neighbor limit of 3 is enforced.
> - **Envelope functions:** From a theoretical perspective, the radial basis functions used in graph neural network machine learning potentials are not twice continuously differentiable due to the finite radius cutoff in graph construction, which causes a step change at the cutoff radius. The envelope function theoretically eliminates this issue.
> - **Number of radial basis functions:** While empirically reducing the number of radial basis functions helps the PES to vary smoothly and improves model conservation properties, due to the flexibility of neural networks, it does not theoretically enforce these properties.
>
>
> We look forward to further discussions if you have additional questions or suggestions. Thank you again for your valuable input.

---

### Official Review · Reviewer_kXc9 · 2025-03-13

**Overall Recommendation:** 4

**Summary:**

This work investigates failure cases of machine learning interatomic potentials (MLIPs) in actual MD simulations. From these insights, the authors draw actionable improvements to MLIP that they implement in their eSEN model. eSEN shows promise in being more accurate on hold-out test sets as well as in preserving energy in MD simulations. This work questions many common design choices that led to reduced test set MAEs but unstable MD simulations.

## After the rebuttal
I stay with my initial judgment. I find this work a pleasant read and well executed. I recommend acceptance.

**Claims And Evidence:**

The authors claim that MLIP should be
1. conservative vector fields.
2. bounded in their derivatives.
3. smooth.

The authors support these claims very well with the implementation of MD simulators, various ablation studies, and empirical evidence.

**Essential References Not Discussed:**

-

**Experimental Designs Or Analyses:**

The experiments are sound and well-analyzed, with meaningful conclusions. However, statistical fluctuations would, by repeating experiments multiple times, benefit communication.

**Methods And Evaluation Criteria:**

The authors offer a broad range of benchmark datasets and include various metrics beyond simple MAE on energies and force to accurately paint a picture. Further, MD simulations are performed with each model to judge its practical usefulness.

**Other Comments Or Suggestions:**

-

**Other Strengths And Weaknesses:**

The paper is very well written, thoroughly investigates failure cases of modern force fields, and draws reasonable conclusions. I enjoyed reviewing this work and am confident that it greatly aids the current field of molecular force fields and the debate on the impact of Euclidean symmetries.

Its main downside is the limited originality in its technical and theoretical contribution. Further, error bars for different trainings would greatly help in indicating the stability of results.

**Questions For Authors:**

-

**Relation To Broader Scientific Literature:**

This work finds that common choices for improving hold-out test set accuracies in machine-learning force fields lead to unphysical behavior in MD simulations. Further, the authors provide clear and concrete guidelines on what properties MLFF should fulfill to yield accurate MD simulations and low test set errors. This work enriches the literature with a fresh evaluation metrics through MD simulation that also correlate well with existing metrics.

**Theoretical Claims:**

The paper makes no theoretical claims.

---

> ### Author Rebuttal · Authors · 2025-03-27
>
> We thank reviewer kXc9 for the helpful feedback. We address each of the reviewer’s comments below.
>
> > Its main downside is the limited originality in its technical and theoretical contribution.
>
> Regarding the originality of our technical and theoretical contributions, we would like to highlight that while energy-conservation-related design choices have been explored in previous studies, our work systematically investigates the impact of each individual design choice. This, to our knowledge, has not been comprehensively addressed before. Our paper's originality lies in (1) elucidating these effects; (2) demonstrating state-of-the-art results when these choices are integrated with the novel eSEN architecture and the direct-force pretraining strategy; (3) we present a novel finding on the correlation between test-set error and physical property prediction performance for models that satisfy the conservation test; (4) we propose the MD conservation test that is critical in establishing the correlation between test errors and downstream predictions.
>
> We believe these findings represent a significant contribution to the community and that they will impact model development practices in the field moving forward.
>
> > Further, error bars for different trainings would greatly help in indicating the stability of results.
>
> We appreciate your suggestion regarding error bars for different training runs. To quantify the variation, we trained 2-layer eSEN models on the MPTrj dataset using 3 different seeds for 50 epochs, with loss coefficients E: 1/F: 10/S: 100. The validation set errors and standard deviations are as below:
> | Task | MAE |
> | -- | -- |
> | Energy (meV/atom)      | 19.67 ± 0.23        |
> | Forces (meV/Å)         | 43.85 ± 0.058       |
> | Stress (meV/Å³ atom)   | 0.16 ± 0.00038      |
>
> We find the results to be highly stable across random seeds. We will incorporate this in our revised manuscript.
>
> We look forward to further discussions if you have additional questions or suggestions. Thank you again for your valuable input.

---

### Official Review · Reviewer_XYRs · 2025-03-14

**Overall Recommendation:** 4

**Summary:**

This paper presents eSEN, a machine learning interatomic potential (MLIP) model designed for accurate and energy-conserving molecular dynamics (MD) simulations and physical property predictions. The study identifies key factors that impact an MLIP’s ability to generalize well to physical property prediction tasks, such as ensuring conservative force predictions and maintaining a smoothly varying potential energy surface (PES). The proposed eSEN model achieves state-of-the-art results on a range of benchmarks, including materials stability prediction, thermal conductivity prediction, and phonon calculations. By establishing a correlation between test errors and physical property prediction performance in energy-conserving models, the authors offer insights into improving the reliability of MLIPs.

**Claims And Evidence:**

The paper claims that energy conservation in MD simulations leads to improved correlation between test errors and downstream physical property predictions. This claim is well-supported by experimental results.

**Essential References Not Discussed:**

The paper provides a comprehensive literature review.

**Experimental Designs Or Analyses:**

The experimental design is well-structured, with clear comparisons between eSEN and existing MLIPs on a representative set of benchmarks. The ablation studies systematically evaluate the impact of various architectural decisions, such as representation discretization, neighbor selection, and envelope functions. The results support the paper’s hypotheses, with energy-conserving models consistently outperforming non-conservative alternatives. However, further exploration of the impact of different hyperparameter choices could strengthen the robustness of these conclusions.

**Methods And Evaluation Criteria:**

The proposed methods and evaluation criteria align well with the problem of developing reliable MLIPs. The authors test their models on a representative range of material property prediction benchmarks including Matbench-Discovery, MDR Phonon benchmark, and SPICE-MACE-OFF.

**Other Comments Or Suggestions:**

1. I would suggest the authors provide some discussion about computational efficiency.

**Other Strengths And Weaknesses:**

Strengths:

1. The experiments in this paper are comprehensive.

2. The proposed eSEN achieves SOTA results on multiple benchmarks, demonstrating strong empirical performance.

Weaknesses:

1. The design choices of the eSEN model are all from existing works.

2. The authors do not provide an empirical study from the perspective of efficiency.

**Questions For Authors:**

1. How is the proposed eSEN evaluated in Table 2? Is it submitted to the Matbench Leaderboard (https://matbench-discovery.materialsproject.org/) for evaluation?

**Relation To Broader Scientific Literature:**

In the field of MLIP, there has been a debate about training conservative (physical but computationally expensive) or non-conservative force fields (computationally efficient). This submission provides great evidence for conservative force fields from the perspective of generalization performance, i.e., the test-set energy error of conservative force fields correlates better with other physical properties.

**Theoretical Claims:**

The theoretical claim of the submission refers to Hairer et al. 2003.

---

> ### Author Rebuttal · Authors · 2025-03-27
>
> We thank reviewer XYRs for the helpful feedback. We address each of the reviewer’s comments below.
>
> > The design choices of the eSEN model are all from existing works.
>
> Regarding the originality of our technical contributions, we would like to highlight that while energy-conservation-related design choices have been explored in previous studies, our work systematically investigates the impact of each individual design choice. This, to our knowledge, has not been comprehensively addressed before. Our paper's originality lies in (1) elucidating these effects; (2) demonstrating state-of-the-art results when these choices are integrated with the novel eSEN architecture and the direct-force pretraining strategy; (3) we present a novel finding on the correlation between test-set error and physical property prediction performance for models that satisfy the conservation test; (4) we propose the MD conservation test that is critical in establishing the correlation between test errors and downstream predictions.
>
> We believe these findings represent a significant contribution to the community and that they will impact model development practices in the field moving forward.
>
> > The authors do not provide an empirical study from the perspective of efficiency.
>
> We would like to respectfully point out that we provided a brief empirical study on the efficiency of eSEN in Appendix C. We found eSEN to have comparable efficiency to existing equivariant models while being more accurate. A 2-layer eSEN model with 3.2M parameters can simulate around 0.8 million steps per day for a periodic system of 216 atoms on a single NVIDIA A100 GPU. We will highlight this result in the main paper in the next revision.
>
> > How is the proposed eSEN evaluated in Table 2? Is it submitted to the Matbench Leaderboard (https://matbench-discovery.materialsproject.org/) for evaluation?
>
>
> We confirm that eSEN was evaluated using the same dataset and metrics as those provided in the Matbench Leaderboard, as referenced in the link you shared. The results have been submitted and verified by the benchmark maintainer, ensuring their accuracy and reliability. The slight difference in the $\kappa_{\mathrm{SRME}}$ metric at the benchmark site is due to a very recent minor update to the Matbench-Discovery evaluation protocol.
>
> While the results of eSEN are on the Matbench-Discovery Leaderboard, we have refrained from directly linking our submission to maintain the integrity of the double-blind review process. We appreciate your understanding in this matter.
>
>
> We look forward to further discussions if you have additional questions or suggestions. Thank you again for your valuable input.

---

### Decision · Program_Chairs · 2025-05-01

**Decision:**

Accept (oral)

**Comment:**

The submission kicks off a serious endeavor to push forward the known but insufficiently attended issue of the quality of MLIP for practical downstream utilities, e.g., MD simulation. The authors noticed that for energy conservation, it is not sufficient to only modeling a gradient field, but also requires smooth operations in an MLIP model, such as discretization of spherical harmonics representations, maximum neighbor limit, and envelope functions. The work conducted a comprehensive study on the effect of these design choices under the consideration of stable MD simulation and downstream physical property prediction. The eSEN model is then proposed to fulfill the requirements based on the insights, which surpasses competitive models on physical property prediction in public benchmarks. Overall, the submission makes a timely and informative contribution to the community, in terms of the emphasis on the problem, approaches to investigate it, observations from the systematic empirical study, and the resulting model useful for challenging downstream tasks.

All reviewers expressed the appreciation of the contribution, and rated quite supportive scores. Some reviewers brought up further queries (empirical efficiency study by Reviewer XYRs, statistical significance by Reviewer kXc9, effect of DeNS by Reviewer yR9i). The authors replied to the questions and queries responsibly.